# Identification and Functional Analysis of the Caffeic Acid O-Methyltransferase (COMT) Gene Family in Rice (*Oryza sativa* L.)

**DOI:** 10.3390/ijms23158491

**Published:** 2022-07-31

**Authors:** Shaoming Liang, Shanbin Xu, Di Qu, Luomiao Yang, Jingguo Wang, Hualong Liu, Wei Xin, Detang Zou, Hongliang Zheng

**Affiliations:** Key Laboratory of Germplasm Enhancement, Physiology and Ecology of Food Crops in Cold Region, Department of Agriculture, Northeast Agricultural University, Harbin 150030, China; leung0415@163.com (S.L.); 18713599521@126.com (S.X.); qudi1997@163.com (D.Q.); yaochang616@163.com (L.Y.); 55190292@163.com (J.W.); liuhualongneau@163.com (H.L.); xinweineau@163.com (W.X.)

**Keywords:** rice, COMT, lignin, SNP, abiotic stress, expression patterns

## Abstract

Caffeic acid O-methyltransferase (COMT) is one of the core enzymes involved in lignin synthesis. However, there is no systematic study on the rice COMT gene family. We identified 33 COMT genes containing the methyltransferase-2 domain in the rice genome using bioinformatic methods and divided them into Group I (a and b) and Group II. Motifs, conserved domains, gene structure and SNPs density are related to the classification of *OsCOMTs*. The tandem phenomenon plays a key role in the expansion of *OsCOMTs*. The expression levels of fourteen and thirteen *OsCOMTs* increased or decreased under salt stress and drought stress, respectively. *OsCOMTs* showed higher expression levels in the stem. The lignin content of rice was measured in five stages; combined with the expression analysis of *OsCOMTs* and multiple sequence alignment, we found that *OsCOMT8*, *OsCOMT9* and *OsCOMT15* play a key role in the synthesis of lignin. Targeted miRNAs and gene ontology annotation revealed that *OsCOMTs* were involved in abiotic stress responses. Our study contributes to the analysis of the biological function of *OsCOMTs*, which may provide information for future rice breeding and editing of the rice genome.

## 1. Introduction

Rice (*Oryza sativa* L.) is an important food crop that requires a suitable environment for growth. Abiotic stress significantly impacts reproductive growth at different times, and salt stress, alkali stress, and drought stress can cause a drastic decrease in rice yield and quality [1,2,3]. Lignin is an important renewable resource in nature [4]. Previous studies have shown that the content of lignin can affect the lodging resistance and disease resistance of plants, but with further research, it has been found that lignin can also affect the abiotic resistance of plants [5].

Lignin is rich in aromatic rings, which is a characteristic of aromatic polymers. Lignin consists of a group of monomers associated with phenyl propane units; the main monomers are coumaryl alcohol, coniferyl alcohol, and sinapyl alcohol [6]. Based on monomer composition, lignin is divided into p-hydroxyphenyl lignin (H-lignin), guaiacyl lignin (G-lignin), and syringyl lignin (S-lignin) [7]. Different plants have different types of lignin, with dicotyledons mainly dominated by G- and S-lignin. Monocotyledons contain all types of lignin, with relatively less H-lignin content, whereas ferns and gymnosperms are mainly dominated by G-lignin [8,9]. Lignin improves plant stress resistance. The level of lignin content can affect the metabolic pathways [10], salt tolerance [11], and drought tolerance [12] of plants.

The biosynthesis of lignin is a process in which phenylalanine or tyrosine is gradually converted into lignin monomers under the catalysis of a series of enzymes, and then, it is finally polymerized into lignin. This pathway consists of three parts: the phenylpropane pathway, the specific pathway for lignin synthesis, and the glycosylation transport and polymerization of lignin monomers to lignin [13]. Caffeic acid O-methyltransferase (COMT) is a crucial methylase in the phenylpropane metabolic pathway. COMT has multiple functions, such as catalyzing the acceptance of the methyl group of S-adenosyl L-methionine (SAM or AdoMet) to form ferulic acid and S-adenosyl L-homocysteine (SAH or AdoHcy) [14,15] to regulate lignin synthesis. COMT also catalyzes the methylation of caffeic acid, 5-hydroxypinobanksyl aldehyde, and 5-hydroxypinobanksyl alcohol to generate ferulic acid, mustard aldehyde, and mustard alcohol, respectively, which are involved in the methylation reactions of mustard alcohol (S unit) and pinobanksyl alcohol (G unit) synthesis, and they play a decisive role in the composition of different types of monolignans [16]. COMT also catalyzes the production of melatonin from N-acetyl-5-hydroxytryptamine [17], and the level of melatonin content affects plant drought tolerance [18] and salt tolerance [19]. The first COMT gene was cloned in *Ligusticum chuanxiong*, and it was concluded that the COMT gene positively regulates the lignin content [20]. In maize, the inhibition of COMT gene expression decreased the total lignin content and S unit/G unit ratio [21]. Overexpression of the COMT gene increased drought tolerance in transgenic *Triticum aestivum* [22]. There are 92 COMT genes in *Vaccinium*, and it has been reported that *VcCOMT40* and *VcCOMT92* play important roles in lignin synthesis [23]. There are 42 COMT genes in *Brassica napus*, and they have been reported to inhibit *BnCOMT1s* expression to enhance lignin content under drought stress [24]. There are 55 COMT genes in *Glycine max*, and the expression levels of *GmCOMTs* under drought and salt stress have been analyzed [25]. In *Solanum lycopersicum*, drought stress and salt stress increased the expression of *SiCOMT1* [26]. In rice, *OsCOMT1* has been reported to affect melatonin content by affecting the 5-methylation of 5-hydroxyferulic acid (5-HFA) and 5-hydroxyconiferaldehyde (5-HCAld) [27]. *OsCAD2* and *OsCOMT15* are involved in lignin monomer synthesis [28]. *OsCOMT3* expression is elevated during nematode infestation in *O. sativa* [29]. Therefore, COMT genes are extremely important for plant resistance to abiotic stress.

In this study, we identified 33 COMT genes in *O. sativa* using bioinformatic tools. We analyzed their chromosome distribution, phylogeny, gene synteny, gene structure, motif composition, cis-acting elements, Go enrichment, tissue expression specificity, expression patterns under abiotic stress, lignin synthesis gene and target miRNAs. Our study focused on the structure and function of *O. sativa* COMT genes. There are relatively few studies on the COMT gene in *O. sativa*. Therefore, our study provides new ideas for future research, especially the molecular mechanism under abiotic stress in rice and the mechanism of lignin synthesis. 

## 2. Results

### 2.1. Physicochemical Properties, Chromosomal Distribution and Synteny Analysis of OsCOMTs

We identified 33 COMT genes from the rice genome. The amino acid length of *OsCOMTs* ranged from 128 (*OsCOMT33*) to 604 (*OsCOMT21*), and the molecular weight (MW) ranged from 14.25 (*OsCOMT33*) to 66.48 *(OsCOMT21*) kDa. *OsCOMTs* predicted PI (isoelectric points) varied from 5.09 (*OsCOMT23*) to 6.46 (*OsCOMT20*). *O. sativa* COMT genes were localized in the cytoplasmic (21 *OsCOMTs*), chloroplast (nine *OsCOMTs*), and mitochondrial (two *OsCOMTs*) (Appendix A). Based on their order on the *O. sativa* chromosome, the COMT genes were named *OsCOMT1*-*OsCOMT33*. The distribution of COMT genes on the chromosomes was uneven and irregular but was mainly concentrated on chromosome 11 (Chr11) and chromosome 12 (Chr12) (Figure 1). Except for chromosome 3 (Chr3) and chromosome 10 (Chr10), all the chromosomes had at least one COMT gene. Both Chr11 and Chr12 contained the most COMT genes (seven genes), whereas Chr1 and Chr2 contained only one gene. 

We analyze the duplication types of *OsCOMTs*. We found five tandem duplication (TD) gene pairs (Figure 1) (*OsCOMT5*/*OsCOMT6*, *OsCOMT8*/*OsCOMT9*, *OsCOMT23*/*OsCOMT24*, *OsCOMT24*/*OsCOMT25*, and *OsCOMT30*/*OsCOMT31*) on four chromosomes (Chr4, Chr5, Chr11, and Chr12). One whole-genome duplication (WGD) event containing two *OsCOMTs* (*OsCOMT1*/*OsCOMT9*) was identified. The *OsCOMTs* with WGD are located on Chr1 and 5 (Figure 1). The results showed the expansion of *OsCOMTs* was mainly achieved through tandem duplication events. The evolutionary selection pressure in *OsCOMTs* differentiation and the Ka/Ks ratio was <1 (Appendix A). The results showed that *OsCOMTs* were subjected to purification during the evolutionary process. 

To investigate the homology of COMT family members among monocots, we produced a collinear map of COMT family genes for six plants, including five monocots (*Oryza sativa*, *Brachypodium distachyon*, *Zea mays*, *Glycine max and Hordeum vulgare*) and one dicot (*Arabidopsis thaliana*). Thirteen colinear gene pairs were identified (Figure 2): *Zm00001eb190920*:*KQK01446*, *Zm00001eb172420*:*KQJ95323*, *Zm00001eb292840*:*KQK05395*, *Zm00001eb353110*:*KQK05395*, *Zm00001eb292840*:*KQK05393*, *Zm00001eb190920*:*OsCOMT2*, *Zm00001eb172420*:*OsCOMT15*, *Zm00001eb292840*:*OsCOMT9*, *Zm00001eb306700*:*OsCOMT13*, *Zm00001eb353110*:*OsCOMT9*, *KRH20371*:*OsCOMT9*, *OsCOMT2*:*HORVU-MOREX-r36-HG0631640* and *OsCOMT9*:*HORVU-MOREX-r3-1HG0077500*. Continuous colinear gene pairs exist in *Oryza sativa*, *Brachypodium distachyon*, *Zea mays*, *Glycine max*, and *Hordeum vulgare.* Before species differentiation, these genes may have been formed from the above-mentioned homologous genes. *OsCOMT9* was found in all four colinear gene pairs. This suggests that *OsCOMT9* plays an important role in the expansion and evolution of the COMT family.

### 2.2. Phylogenetic Analysis, Motif, Conserved Domain and Gene Structure of COMT Genes in O. sativa

We constructed a maximum likelihood (ML) phylogenetic tree using 109 protein sequences of COMT genes from six plant species including five monocots: *Oryza sativa*, *Brachypodium distachyon*, *Zea mays*, *Glycine max*, and *Hordeum vulgare* and a dicot of *Arabidopsis thaliana*. All the sequences clustered into two groups (Figure 3 and Figure 4). There were 24 and 9 COMT proteins in Groups I and II, respectively. Comparing the five types of monocots, the COMT proteins in the dicot of *Arabidopsis thaliana* clustered in Group II. Although the COMT proteins of the five monocots clustered in different subgroups, the degree of COMT protein clustering was higher in different species within the same subgroup. The results showed that COMT genes were highly divergent between monocot and dicot, while COMT genes were conserved among monocots. 

We analyzed the differences in the protein sequences of the COMT genes using the MEME program. We identified 12 motifs associated with gene classification in the protein sequences of the COMT genes (Appendix A). These motifs are associated with gene classification. All *OsCOMTs* had motif 8 except *OsCOMT33*. The *OsCOMTs* in Group II all contained motif 12, while the *OsCOMTs* in Groups Ia and Ib contained motif 10, except for *OsCOMT32*. In the conserved domain analysis, there was a methyltransf_2 domain (including a SAM/SAH binding pocket and a substrate-binding site) in all *OsCOMTs* (Figure 5). All *OsCOMTs* also contained an N-terminal domain named dimerization except for *OsCOMT4* and *OsCOMT33* [30]. The SAM/SAH binding region was highly conserved, whereas substrate-binding sites were specific for different groups of proteins. 

By demonstrating the gene structure of *O. sativa* COMT through GSDS2.0 [31], we found that its exons are distributed differently in Groups I and II (Figure 5). In Group Ia, all *OsCOMTs* contained two exons. All *OsCOMTs* in Group Ib contained one or two exons. In Group II, all *OsCOMTs* contained two or more exons. *OsCOMT11* and *OsCOMT33* do not contain introns. Four *OsCOMTs* (*OsCOMT10*, *OsCOMT11*, *OsCOMT33*, and *OsCOMT32*) contained one exon, and two *OsCOMTs* (*OsCOMT3* and *OsCOMT6*) contained three exons, four *OsCOMTs* (*OsCOMT2*, *OsCOMT4*, *OsCOMT5*, and *OsCOMT29*) contained four exons, and the other *OsCOMTs* contained two exons. These results indicated that *O. sativa* COMT genes in different phylogenetic branches may have different biological functions.

### 2.3. Cis-Acting Elements, Sequence Variation and Gene Ontology Annotation of OsCOMTs 

Cis-acting elements in plants can bind to multiple transcription factors that influence gene expression. To investigate the function of the *OsCOMTs*, we obtained and submitted a 1500 bp gene sequence upstream of *OsCOMTs* and classified them into three types of cis-acting elements according to their functions: Abiotic stress, Hormone response and Plant growth and regulation (Figure 6A–C). In terms of the number of cis-acting elements, *OsCOMTs* are widely involved in plant responses to abiotic stresses, the hormonal regulation of plants and plant growth and development. In terms of abiotic stress, MBS (drought-inducible) [32] were the most cis-acting. This implies that *OsCOMTs* may be involved in drought stress response. In addition, there are anaerobic induction elements (ARE) [33] and cold induction elements (LTR) [34]. ABRE (abscisic acid responsive) [35] were the most abundant elements in hormone regulation, which were followed by TGACG-motif (MeJA responsive) [36]. G-box (light response) elements [37] were the most abundant elements in plant growth and regulation. The results show that *OsCOMTs* can be widely involved in various regulation in plants, but there is no significant difference in the distribution of elements in Group I and Group II.

According to the resequencing results of 295 japonica rice varieties by our group [38], we found 236 SNPs in *OsCOMTs*. On average, there are 7.18 SNPs per OsCOMT. The distribution of its SNPs are related to gene grouping. The SNP densities in Group Ia, Group Ib and Group II were 0.67, 9.73 and 14.67. In Group Ia, only *OsCOMT13*, *OsCOMT16* and *OsCOMT24* have SNPs. There are no SNPs in *OsCOMT8*, *OsCOMT9* and *OsCOMT10* in Group Ib. In Group II, only *OsCOMT25* has no SNP (Appendix A). This indicates that the *OsCOMTs* in Group Ia are highly conserved among different rice genotypes. This difference in SNP density may be related to the function of *OsCOMTs*.

To further investigate the functions of *OsCOMTs*, Go enrichment was performed on all *OsCOMTs*. Their functions include S-adenosylmethionine-dependent methyltransferase activity (31 *OsCOMTs*), protein dimerization activity (29 *OsCOMTs*), O-methyltransferase activity (33 *OsCOMTs*), methylation (33 *OsCOMTs*), melatonin biosynthetic process (2 *OsCOMTs*), aromatic compound biosynthetic process (31 *OsCOMTs*) and acetylserotonin O-methyltransferase activity (2 *OsCOMTs*) (Figure 5D).

### 2.4. Expression Profiling of OsCOMTs

The current research on the involvement of COMT genes in the regulation of plant stress resistance focuses on salt stress and drought stress, so we studied the expression of *OsCOMTs* under these two stresses by qRT-PCR. Under salt stress, the expression levels of seven *OsCOMTs* were decreased and increased, respectively (Figure 6A). *OsCOMT20* and *OsCOMT31* were strongly induced by salt stress. Under drought stress, the expression of eight *OsCOMTs* increased and the expression of six *OsCOMTs* decreased (Figure 6B). *OsCOMT2*, *OsCOMT5*, *OsCOMT16* and *OsCOMT21* were strongly induced by drought stress. The six *OsCOMTs* were induced to express under both stresses, and their expression patterns were different. The expression levels of *OsCOMT7* and *OsCOMT20* were increased under salt stress but decreased under salt stress.

To investigate the tissue expression specificity of COMT genes, we studied the expression level of *OsCOMTs* in leaf, emerging_inflorescence, early_inflorescence, pistil, 5_days_seed, 10_days_seed, embryo_25_days_after_pollination, 25_Days_endosperm, endosperm_25_days_after_pollination, stem, root, and anthers (Appendix A). In total, 12 *OsCOMTs* were highly expressed in leaf, 11 *OsCOMTs* were highly expressed in the stem, 6 *OsCOMTs* were highly expressed in emerging_inflorescence, and 4 *OsCOMTs* were highly expressed in roots. *OsCOMTs* induced by salt and drought stress were detected in at least one tissue, so we further verified by qRT-PCR(Figure 6C). The results showed that the expression levels of *OsCOMTs* were higher in stems, leaves and seeds, especially stems. The expression level was lower in roots; only *OsCOMT14* and *OsCOMT28* were expressed at higher levels in roots. This indicates that in the growth and development of rice, *OsCOMTs* play their roles in leaves, seeds and stems.

### 2.5. Spatial Expression of OsCOMTs in Stem

We first analyzed the expression patterns of *OsCOMTs* in five stages of rice by qRT-PCR. We simultaneously measured the lignin content in rice stems in five stages (Appendix A). The lignin content increased gradually from S1 to S5, slowly from S1 to S2 and S3 to S4, increased more from S2 to S3, and reached a peak at S5. The expression of *OsCOMTs* were higher in S5 (Figure 7A). The expression of *OsCOMT8*, *OsCOMT9* and *OsCOMT15* was similar to the change of lignin content, but the expression of *OsCOMT9* increased more from S2 to S3 (Figure 7B). Based on the multiple sequence alignment with known COMT genes (Figure 8), we found that these three COMT genes have the same substrate binding site to catalyze 5-OH coniferalde-hyde and caffeic acid, suggesting that *OsCOMT8*, *OsCOMT9* and *OsCOMT15* play an important role in lignin synthesis [39].

### 2.6. Network of Protein Interaction and miRNAs

In total, 31 *OsCOMTs* were predicted to have potential targeting relationships with 681 miRNAs from 200 families (Appendix A). miR5071, miR2927, miR1864 and miR5809 target 14, 13, 10, and 9 *OsCOMTs*, respectively. This suggests that these four miRNAs play important roles for *OsCOMTs* (Figure 9).

## 3. Discussion

In our study, we identified 33 *O. sativa* COMT genes from the whole genome of *O. sativa* and named them *OsCOMT1*–*OsCOMT33* according to their positions on the chromosome. Based on gene structure, conserved motifs, and phylogenetic analysis, we divided the COMT genes into two groups and named them Group I (a and b) and Group II. This is consistent with the evolutionary analyses of blueberries [23] and soybeans [25]. There was a higher similarity of conserved motifs and gene structures in the same branch. In Group Ia, *OsCOMTs* contained two exons, *OsCOMTs* in Group Ib contained one or two exons, and *OsCOMTs* in Group II contained two or more exons. Compared to Group I, the *OsCOMTs* in Group II contained more instances of motif 12 (VDRMLR). Some residues of motif 2 (DVGGG), motif 9 (DLPHV), and motif 5 (GDMF) are the SAM/SAH binding sites [40]. We also constructed a phylogenetic tree consisting the sequence of five monocots and one dicot and discovered that the COMT genes of Arabidopsis were distributed in Group II. This suggests that COMT genes are highly divergent between monocot and dicot and that the evolution of COMT genes in monocots is stable.

Gene duplication has played a key role in the evolution of plant gene families. Five tandem gene pairs and one WGD gene pair were identified using MCScanX in the rice genome. Thus, tandem phenomena play a key role in the evolution of the rice COMT genes. The rice COMT duplication gene pairs were concentrated on four chromosomes. This phenomenon may be due to the loss of some copies of COMT on the 12 rice chromosomes due to environmental factors, some factors received during the evolution of rice, or the presence of some redundant genes with incomplete structural domains. Ka/Ks < 1 in all duplicated gene pairs, suggesting that rice COMT genes underwent purifying selection during evolution. We also investigated the homology of the COMT genes in monocots and identified 13 colinear gene pairs. We found that the COMT gene was relatively conserved during the evolution of monocots. *OsCOMT9* was found in several homologous, WGD, and tandem gene pairs. These results indicate that *OsCOMT9* is a key gene involved in the evolution and expansion of the rice COMT gene family. Cis-acting elements can bind to transcription factors to regulate genes. Therefore, studying the cis-acting elements of genes facilitates gene function analysis. To investigate the function of *OsCOMTs*, we analyzed their cis-acting elements. We classify the cis-acting elements of *OsCOMTs* into three types. Twenty-three *OsCOMTs* contain MBS (drought-inducible) cis-acting elements. Previous studies have shown that the *O. sativa* COMT gene regulates lignin synthesis [28] and that lignin content affects drought resistance in *O. sativa*. Therefore, we suggest that *OsCOMTs* are involved in drought regulation by regulating lignin synthesis. GO enrichment indicated that all *OsCOMTs* were involved in methylation and O-methyltransferase activity. Methylation and O-methyltransferase activity play important roles in the conversion of phenylalanine to lignin [41]. In addition, 94% of *OsCOMTs* are involved in the aromatic compound biosynthetic process and S-adenosylmethionine-dependent methyltransferase activity. The aromatic properties of lignin, which determine the hydrophobicity of the cell wall, are conducive to reducing water loss and thus improving drought tolerance in plants under drought stress [5]. The content of S-adenosylmethionine synthetase in plants can be induced by salt stress, high-temperature stress and drought stress [42,43]. This proves that *OsCOMTs* play an important role in lignin synthesis and resistance to abiotic stress. 

Previous studies have shown that the COMT gene family can be involved in plant abiotic stress responses [25]. We identified fourteen *OsCOMTs* induced by salt stress and thirteen *OsCOMTs* induced by drought stress. The expression of *OsCOMTs* increased or decreased at different time points under abiotic stress, suggesting that COMT genes can be induced by abiotic stress. Some *OsCOMTs* show different expression patterns under different adversities. The expression of *OsCOMT7* and *OsCOMT20* increased under salt stress and decreased under drought stress. Furthermore, we found that the induced *OsCOMTs* were more conserved at the evolutionary level. The induced *OsCOMTs* had an average of 5.95 SNPs, while the 33 *OsCOMTs* had an average of 7.18 SNPs. In conclusion, these results suggest that *OsCOMTs* are involved in the abiotic stress response network. We also found that *OsCOMTs* had significant tissue-specific expression, with *OsCOMTs* showing high expression in stems and very low expression in roots and even some *OsCOMTs* were not expressed. By measuring the lignin content at different growth stages, we found that the lignin content in the rice stem increased gradually. The reason for this phenomenon may be related to lignification. During the development of rice, xylem and phloem undergo lignification, which plays a role in guiding and mechanically supporting the development of rice. The lignification process is the precipitation process of lignin in plants. During the lignification process of plants, lignin penetrates into the cell wall and fills the cell wall framework, thereby enhancing the stiffness of the cell wall and having a significant impact on the mechanical strength of the stem [44]. Combined with multiple sequence alignment and expression analysis of *OsCOMTs* in five developmental stages, *OsCOMT8*, *OsCOMT9* and *OsCOMT15* may play key roles in stem lignin synthesis.

We also analyzed the targeted miRNAs of *OsCOMTs*. miR5071, miR2927, miR1864 and miR5809 target multiple *OsCOMTs*. miR5071 targets *OsMLA10* [45] and is highly expressed in endosperm. miR2927 plays an important role in drought resistance [46]. miR1864 is down-regulated when rice is under arsenic stress [47]. miR5809 is involved in heat stress response [48]. These results indicate that COMT genes may be widely involved in various abiotic stress responses. Our findings contribute to the study of *OsCOMTs*. Future studies need to explore the signaling systems and downstream pathways of COMT genes under abiotic stress.

## 4. Materials and Methods

### 4.1. Identification of COMT Family Members in O. sativa and in Other Plants

The genome database of *O. sativa* was downloaded from EnsemblPlants (https://plants.ensembl.org/index.html accessed on 10 April 2022). To identify rice COMT genes, one signature sequence of *Arabidopsis thaliana* (AT5G54160) and 36 COMT sequences of *Populus tomentosa* were used as a set of queries in a BLASTP. All candidate sequences were scanned with a domain (PF00891) from Pfam. Then, each sequence was confirmed by the SEQUENCE SEARCH in the Pfam website (https://pfam.xfam.org accessed on 10 April 2022), the Conserved Domain Database (CDD) (http://www.ncbi.nlm.nih.gov/Structure/bwrpsb/bwrpsb.cgi accessed on 10 April 2022) and the SMART web server (http://smart.embl.de/ accessed on 10 April 2022). The isoelectric points and molecular weights of *OsCOMTs* proteins were predicted with ExPASy (http://web.expasy.org/protparam/ accessed on 10 April 2022). We also identified COMT genes in *Arabidopsis thaliana, Brachypodium distachyon, Zea mays, Glycine max, and Hordeum vulgare*.

### 4.2. Gene Structure, Conserved Motifs, and Phylogenetic Analysis

We identified 33 candidate rice COMT genes using BLASTP and a protein domain (PF00891). The MEME program (http://meme-suite.org/meme/ accessed on 15 April 2022) was used to analyze the conserved motifs of *OsCOMT* protein sequences, while the additional conserved motifs were predicted by SEQUENCE SEARCH on the Pfam website (http://pfam.Xfam.ori/ accessed on 15 April 2022) and visualized using the Gene Structure View of TBtools (v.1.098745) [49]. A phylogenetic tree of the COMT protein sequences of *Oryza sativa*, *Brachypodium distachyon*, *Zea mays*, *Glycine max*, and *Hordeum vulgare* was constructed using the ML method in MEGA(v.7.0) [50] and embellished by iTOL(v.3.0) [51].

### 4.3. Gene Duplication Analysis

Colinear blocks between genes of different species were analyzed using MCScanX [52]. To analyze the extent of evolutionary selection pressure in *OsCOMTs* differentiation, the ratio of Ka (nonsynonymous) to Ks (synonymous) was calculated for the tandem and segmental gene pairs using TBtools(v.1.098745) [49].

### 4.4. Cis-Acting Elements Analysis and miRNAs

The 1.5 kb sequence upstream of *OsCOMTs* was obtained from EnsemblPlants (https://plants.ensembl.org/index.html accessed on 20 April 2022), and cis-acting elements were analyzed using PlantCARE(http://bioinformatics.psb.ugent.be/webtools/plantcare/html accessed on 20 April 2022) [53]. We used rice miRNA sequences obtained from the miRbase database (https://www.mirbase.org/ accessed on 20 April 2022) to search for candidate targets in the PSRNATTARGE and embellished with Cytoscape [54].

### 4.5. Expression Analysis of COMT Genes in O. sativa

The data for *OsCOMTs* in different tissue were downloaded from the Sequence Read Archive under the accession number SPR008821. The heatmaps were obtained using TBtools [49].

### 4.6. Plant Growth Conditions and Treatments

Xiaobaijingzi (XBJZ) is a drought-tolerant *O. sativa*. The seeds of XBJZ (from the Northeast Agricultural University Rice Research Group, provided by professor Hongliang Zheng) were sterilized with 10% sodium hypochlorite for 30 min, then sown on the medium and incubated. The seeds developed into seedlings two weeks later and were transplanted into Hoagland’s nutrient solution. At the three-leaf stage [55], the rice was exposed to abiotic stress. We added 20% PEG-6000 to the nutrient solution to simulate drought stress, and we added 200 mmol/L NaCl to the nutrient solution to simulate salt stress. The stems were collected after 6 h, 12 h, and 24 h of treatment and then stored at −80 °C. We also collected roots, stems, leaves and seeds under normal conditions and stems at seedling, tillering, booting, heading and grain filling stages of rice. Each experiment was repeated 3 times.

### 4.7. Expression Analysis of OsCOMTs in Rice by qRT-PCR

Primer design (Appendix A) was performed using Primer Premier 5.0, and primer specificity was verified using the BLAST program in NCBI (https://ncbi.nlm.nih.gov accessed on 20 April 2022). Total RNA was extracted using an Ultra-Pure Total RNA Extraction Kit (Hangzhou Sumgen Biotech Co., Ltd., Hangzhou, China) and stored at −80 °C. First-strand cDNA (10 µL) was synthesized according to the instructions for the PrimeScript™ RT Master Mix (Takara Biomedical Technology (Beijing) Co., Ltd., Beijing, China). The internal reference gene was OsSRFP1. qRT-PCR was performed in the LightCycler 96 system software using SYBR green (Vazyme Biotech Co., Ltd., Nanjing, China) and fluorescent dyes. Finally, the expression of *OsCOMTs* was calculated using the 2-∆∆CT method [56].

### 4.8. Lignin Content Analysis

Determination of Klason lignin according to the previous method [24].

## 5. Conclusions

In this study, we identified 33 rice COMT genes. Different groups of *OsCOMTs* have different motifs, gene structures and SNPs densities. COMT genes are highly differentiated between monocot and dicot, and the tandem duplication phenomenon is the main force for the expansion of rice COMT gene family. We analyzed the cis-acting elements of *OsCOMTs* and divided them into three types, indicating that *OsCOMTs* can respond to abiotic stresses and participate in various regulation of rice. Twenty *OsCOMTs* were differentially expressed under different types of abiotic stresses. *OsCOMT8*, *OsCOMT9*, and *OsCOMT15* play critical roles in lignin synthesis by measurement of lignin and qRT-PCR. Through miRNA analysis, *OsCOMTs* can respond to abiotic stress. Our findings provide a solid foundation for understanding the role of *OsCOMTs* as well as their expression patterns and mechanisms regulating stress resistance.

## Figures and Tables

**Figure 1 ijms-23-08491-f001:**
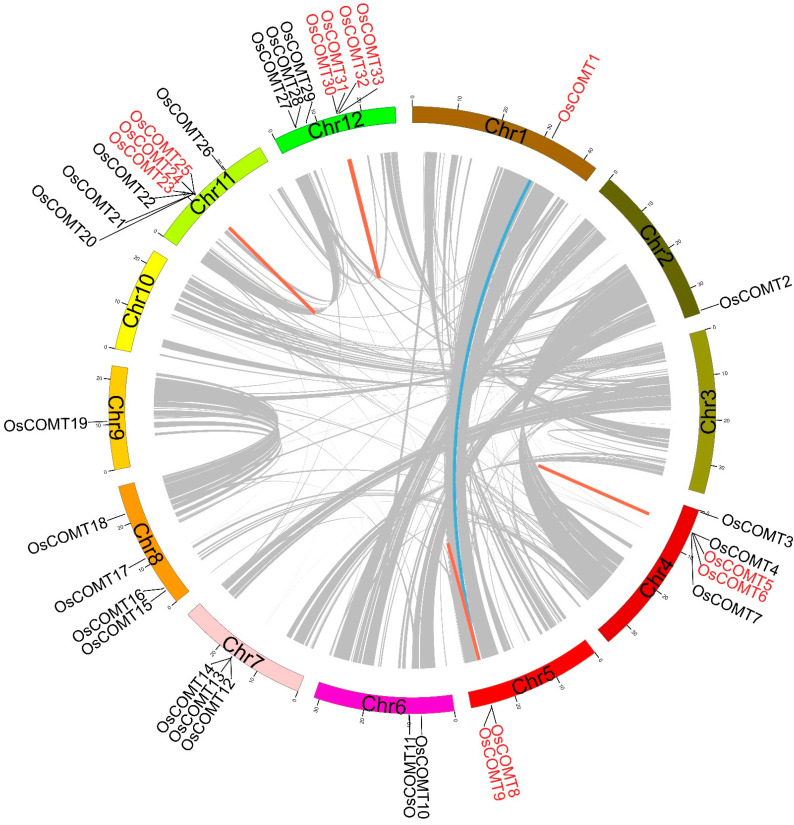
The gene duplication types and chromosomal distribution of *OsCOMTs*. The WGD gene pair is linked with a thick blue line. TD gene pairs are linked with thick orange lines. The duplications genes were marked in red. The scale marked on the chromosome indicates the chromosome length (Mb).

**Figure 2 ijms-23-08491-f002:**
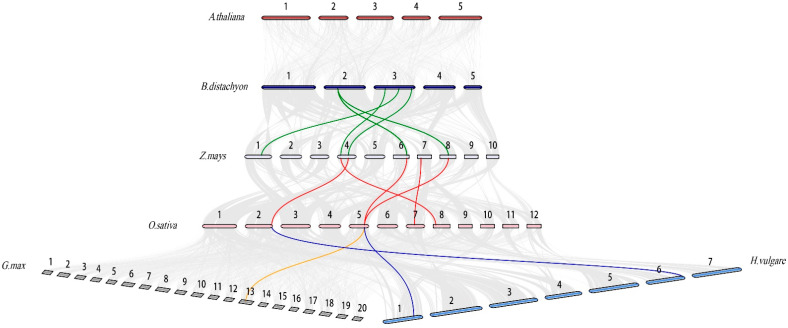
Synteny analysis of *Arabidopsis thaliana*, *Oryza sativa*, *Brachypodium distachyon*, *Zea mays*, *Glycine max*, and *Hordeum vulgare*. The chromosomes of different species are represented by long bars of different colors.

**Figure 3 ijms-23-08491-f003:**
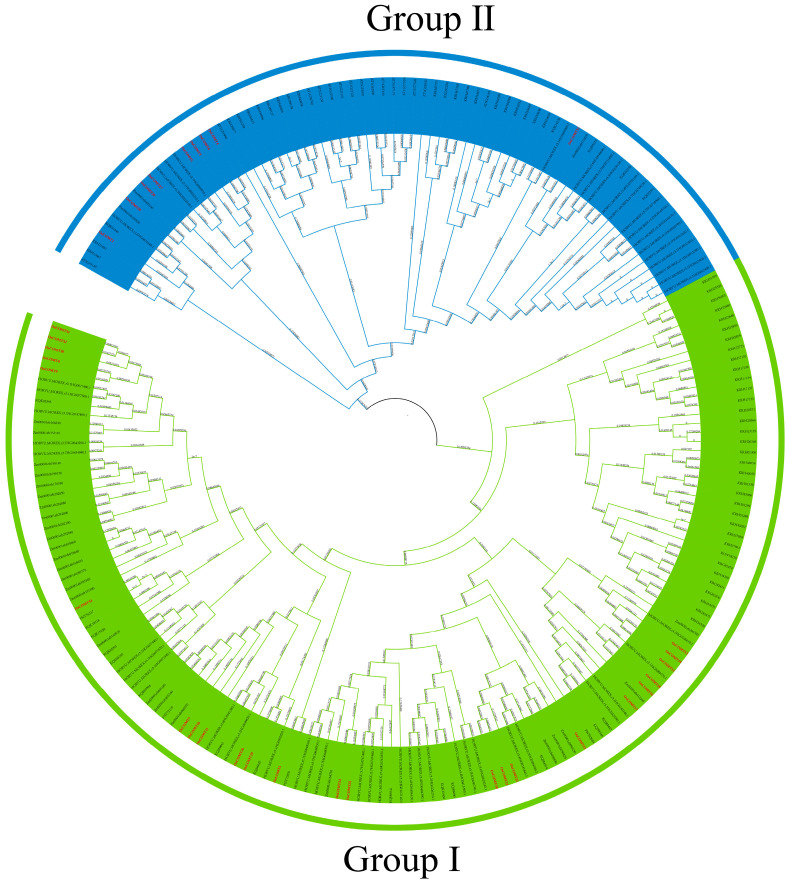
Phylogenetic relationships among 209 COMT proteins in *Arabidopsis thaliana*, *Oryza sativa*, *Brachypodium distachyon*, *Zea mays*, *Glycine max*, and *Hordeum vulgare*. The *OsCOMTs* are marked in red.

**Figure 4 ijms-23-08491-f004:**
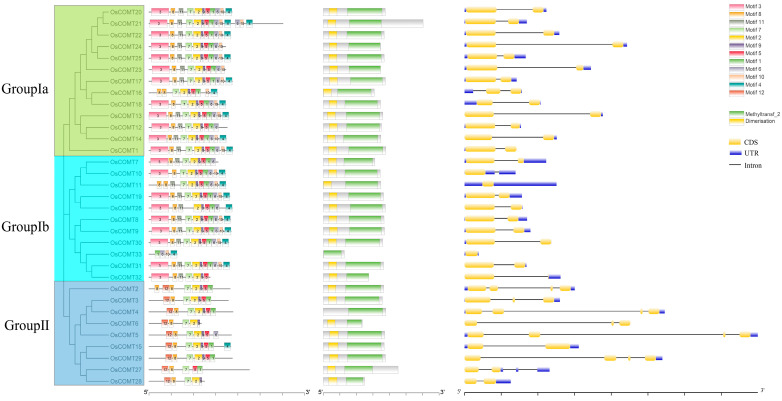
Domain motif and gene structure of *OsCOMTs*. Different colors were used to represent different domain motifs. The yellow round-corner rectangle represents CDS, the blue rectangle represents UTR, and the black line represents introns.

**Figure 5 ijms-23-08491-f005:**
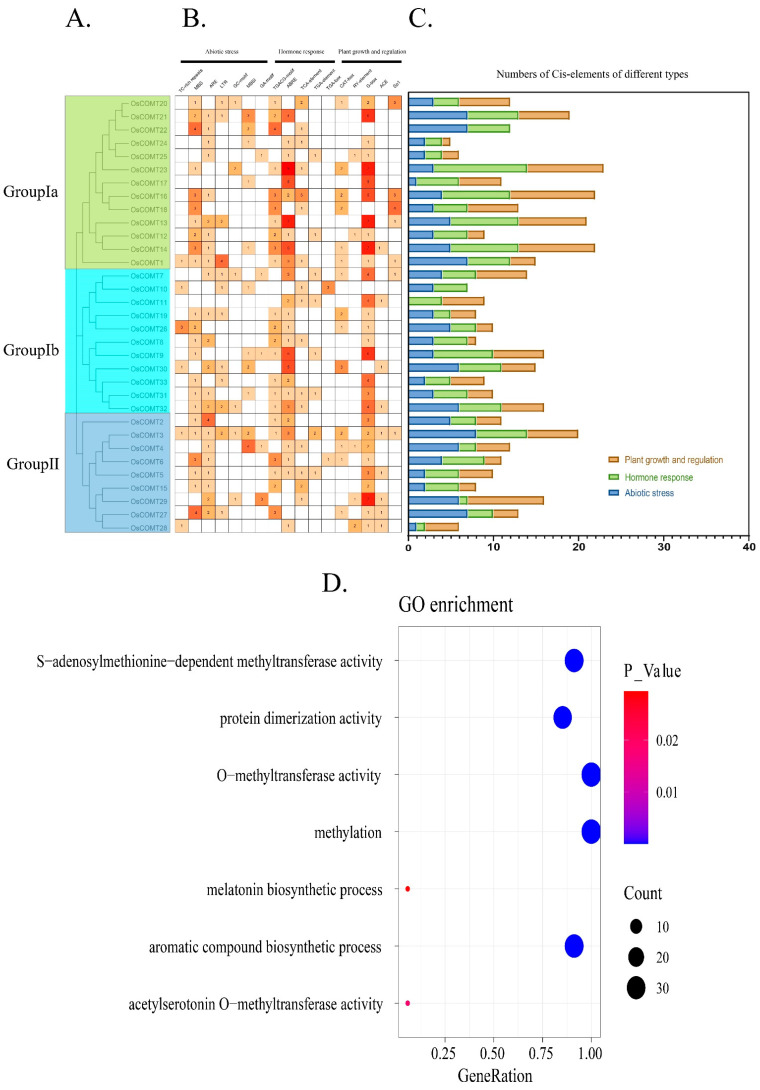
(**A**) Phylogenetic tree of *OsCOMTs*. (**B**) Cis-acting elements heatmap of *OsCOMTs*. (**C**) Histograms of different colors represent different types of elements. (**D**) GO enrichment analysis of *OsCOMTs*.

**Figure 6 ijms-23-08491-f006:**
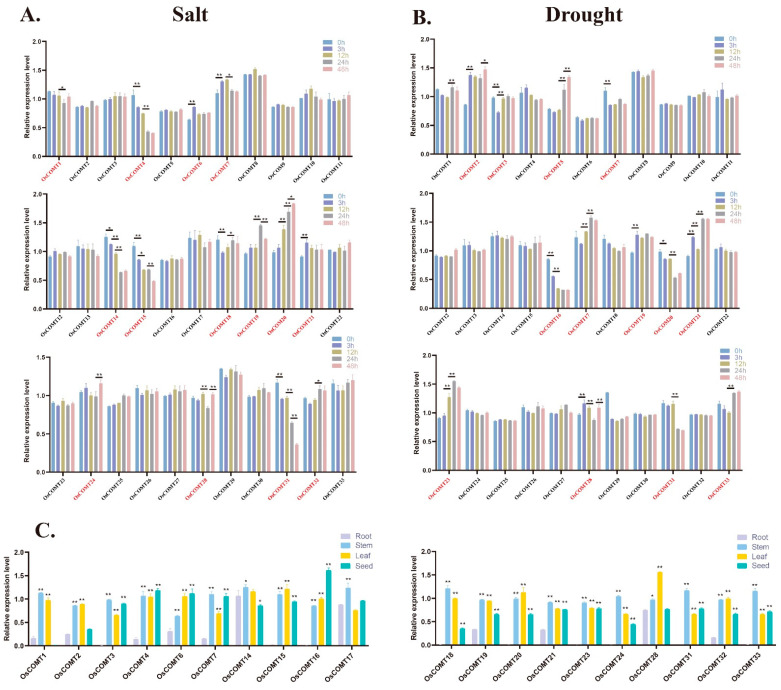
(**A**) Expression levels of differentially expressed *OsCOMTs* under salt stress. (**B**) Expression levels of differentially expressed *OsCOMTs* under drought stress. (**C**) Expression of *OsCOMTs* in seeds, stems, leaves, and roots. Induced *OsCOMTs* are marked in red. Asterisks indicate that the expression of *OsCOMTs* was significantly increased or decreased at different times after treatment. (* *p* < 0.05; ** *p* < 0.01; Student’s *t*-test).

**Figure 7 ijms-23-08491-f007:**
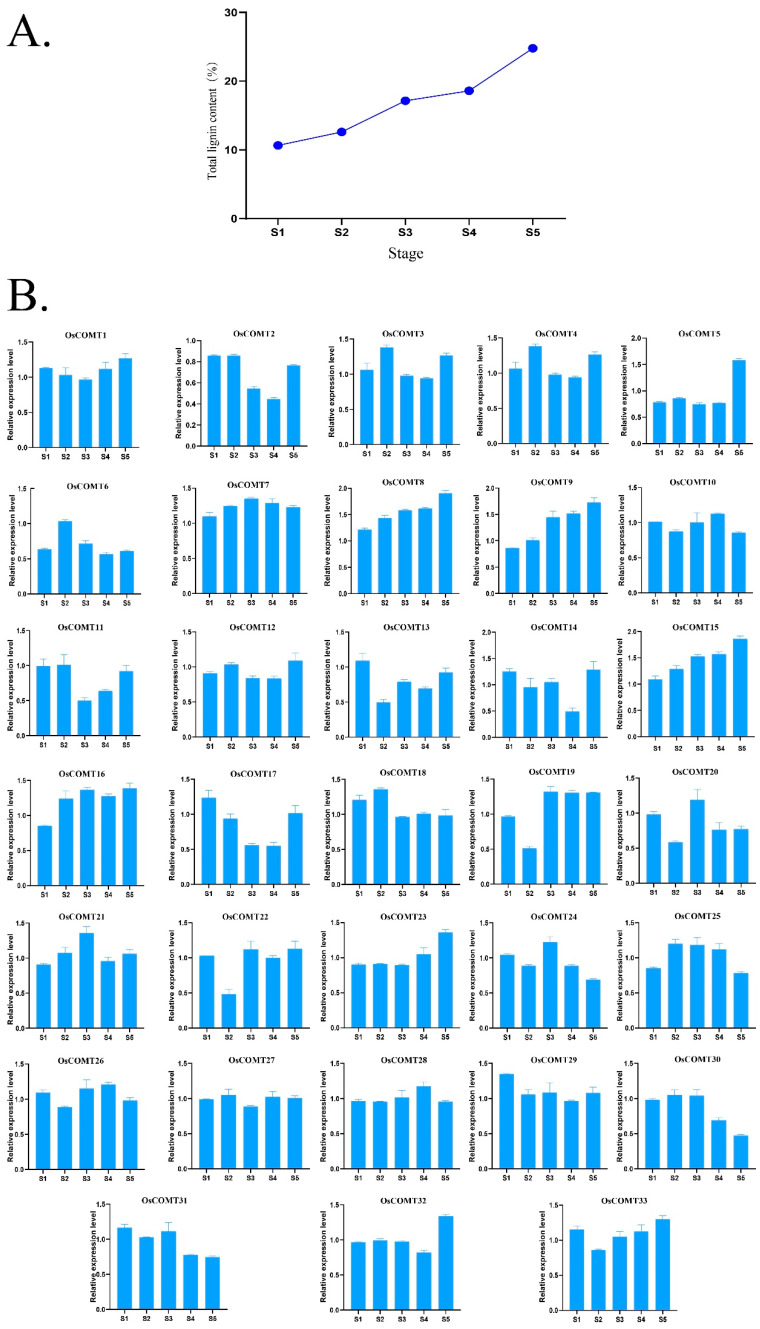
(**A**) Lignin content in five growth stages of rice stem. (**B**) Expression of *OsCOMTs* in rice stem at five different stages. S1: Seedling stage, S2: Tillering stage, S3: Booting stage, S4: Heading stage, S5: Filling stage.

**Figure 8 ijms-23-08491-f008:**
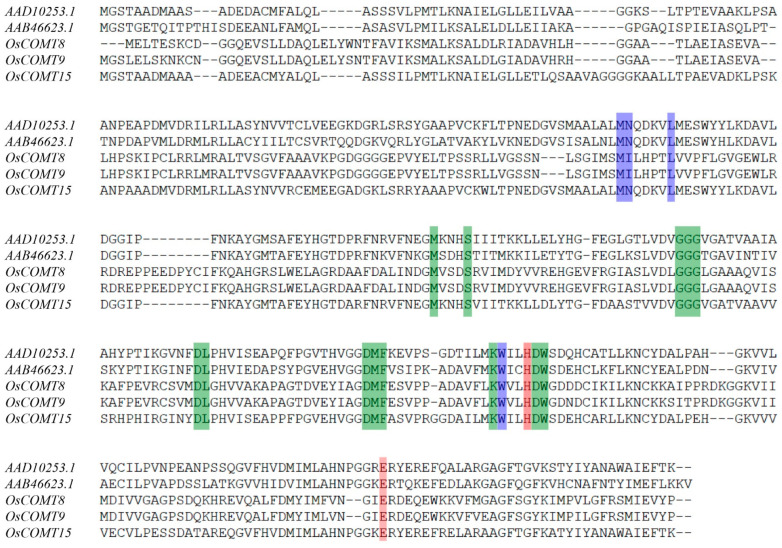
Multiple sequence alignment of *OsCOMT8*, *OsCOMT9* and *OsCOM15* with known COMT genes. Purple: Substrate; Green; SAM binding; Orange: catalytic residues.

**Figure 9 ijms-23-08491-f009:**
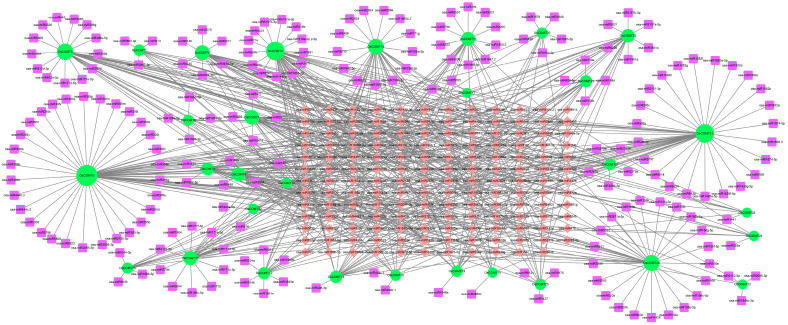
Network diagram of miRNAs targeted with *OsCOMTs*. *OsCOMTs* are marked in green. Single-targeted miRNAs are marked in purple, and multi-targeted miRNAs are marked in pink.

## Data Availability

The datasets generated for this study can be found in SRA, the accession number: SPR008821.

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
