# Peer review of "Identification and Functional Analysis of the Caffeic Acid O-Methyltransferase (COMT) Gene Family in Rice (*Oryza sativa* L.)"

_ijms, 2022, doi:10.3390/ijms23158491_

Round 1
Reviewer 1 Report
Comments to Authors
Interesting article.
2.2 Authors mention 5 graminaceous species - but only 4 are listed (Glycine max/soybean is not graminaceous).
Figure 6c. No statistical analysis?
Figure 7a. No label on X-axis.
Author Response
Response to Reviewer Comments
Dear Reviewer,
Thank you for your reviewer comments concerning our manuscript entitled “Identification and functional analysis of the Caffeic Acid O-Methyltransferase(COMT) gene family in rice (Oryza sativa L.)” (Manuscript ID: ijms-1841348). Those comments are all valuable and very helpful for revising and improving our paper, as well as the important guiding significance to our researches. We have studied comments carefully and have made correction which we hope meet with approval. Revised portion are marked in red in the paper. The main corrections in the paper and the responds to the reviewer’s comments are as follow:
Point 1:Interesting article.
Thank you very much for reviewing this manuscript.
Point 2: 2.2 Authors mention 5 graminaceous species - but only 4 are listed (Glycine max/soybean is not graminaceous).
We are very sorry for our negligence. We have modified 5 graminaceous species to 5 monocots. We have also revised the parts of the article that involve this error.
Page3, line106, 107and109
Page5, line136, 137, 140, 141 and 144
Page15, line283, 284 and 286
Page16, line301 and 302
Page18, line419
Point 3: Figure 6c. No statistical analysis?
We are very sorry for our negligence. We have added statistical analysis.(Figure 6c)
Point 4: Figure 7a. No label on X-axis.
We are very sorry for our negligence. We have added label to the X-axis.
Special thanks to you for your comments. We tried our best to improve the manuscript and made some changes in the manuscript. We appreciate for Editors/Reviewers’ warm work earnestly, and hope that the correction will meet with approval.
Once again, thank you very much for your comments and suggestions.

Reviewer 2 Report
Dear Authors
I am impressed with the work you conducted however there is there is little information needed to improve thos manuscript before its acceptance.
Improvement of conclusion is required and also please increase the figure resolution
Author Response
Response to Reviewer Comments
Dear Reviewer,
Thank you for your reviewer comments concerning our manuscript entitled “Identification and functional analysis of the Caffeic Acid O-Methyltransferase(COMT) gene family in rice (Oryza sativa L.)” (Manuscript ID: ijms-1841348). Those comments are all valuable and very helpful for revising and improving our paper, as well as the important guiding significance to our researches. We have studied comments carefully and have made correction which we hope meet with approval. Revised portion are marked in red in the paper. The main corrections in the paper and the responds to the reviewer’s comments are as follow:
Point 1: I am impressed with the work you conducted however there is there is little information needed to improve thos manuscript before its acceptance.
Thank you very much for reviewing this manuscript.
Point 2: Improvement of conclusion is required.
Thank you very much for your comments. In the revised manuscript, we have supplemented the content of the conclusions section. (line417-422)
Point 3: Please increase the figure resolution.
Thank you very much for your comments. We have increased the resolution of all figures
Special thanks to you for your comments. We tried our best to improve the manuscript and made some changes in the manuscript. We appreciate for Editors/Reviewers’ warm work earnestly, and hope that the correction will meet with approval.
Once again, thank you very much for your comments and suggestions.
